# A tailored cognitive behavioral program for juvenile justice-referred females at risk of substance use and delinquency: A pilot quasi-experimental trial

Sarah C. Walker[1]*, Mylien Duong[1], Christopher Hayes[2], Lucy Berliner[3], Leslie D. Leve[4], David C. Atkins[1], Jerald R. Herting[5], Asia S. Bishop[6], Esteban Valencia[1]¤

1 Department of Psychiatry and Behavioral Sciences, University of Washington, Seattle, Washington, United States of America, 2 Snohomish County Juvenile Court, Everett, Washington, United States of America, 3 Harborview Center for Sexual Assault and Trauma, Seattle, Washington, United States of America, 4 College of Education, University of Oregon, Eugene, Oregon, United States of America, 5 Department of Sociology, University of Washington, Seattle, Washington, United States of America, 6 Department of Social Work, University of Washington, Seattle, Washington, United States of America

¤ Current address: School of Public Health, University of Washington, Seattle, Washington, United States of America
* secwalkr@uw.edu

**Data Availability Statement:** The data has been uploaded to Figshare and can be accessed via

## Abstract

This pilot quasi-experimental trial tested a gender-responsive cognitive behavioral group intervention with 87 court-involved female adolescents (5 juvenile courts) who were at indicated risk for substance use disorder. Participants in the intervention ($n = 57$) received twice weekly group sessions for 10 weeks (20 sessions) focused on building emotional, thought and behavior regulation skills and generalizing these skills to relationally-based scenarios (GOAL: Girls Only Active Learning). Youth in the control condition ($n = 30$) received services as usual, which included non-gender-specific aggression management training, individual counseling and no services. The GOAL program was found to be acceptable to youth and parents and feasible to implement within a juvenile court setting using skilled facilitators. Compared to services as usual, the program significantly and meaningfully reduced self-reported delinquent behavior ($\beta = 0.84$, $p < 0.05$) over 6 months, and exhibited trend level effects for reduced substance use ($\beta = 0.40$, $p = 0.07$). The program had mixed or no effects on family conflict and emotion regulation skills. These findings are discussed in light of treatment mechanisms and gender-responsive services.

## Introduction

Sustained substance use for females is associated with increased risk for HIV infection, violent victimization and downstream risk for birth defects [1–3]. Preventing substance use disorders with effective programs for selective and indicated prevention populations can yield substantial savings in avoided medical, system and human costs and are a good investment for social

(https://figshare.com/articles/GOALEvalPlosone_
1_xls/10022852).

**Funding:** This study was funded by the National
Institute of Drug Abuse (www.drugabuse.gov) to
S.C.W. (R21 DA037455). The funder had no role in
study design, data collection and analysis, decision
to publish, or preparation of the manuscript.

**Competing interests:** S.C.W., L.B. and C.H. are
listed as inventors of the GOAL program under a
University of Washington license and the invention
was reported to the Department of Health and
Human Services on 03/30/2018. No patent is
pending. This does not alter our adherence to
PLOS ONE policies on sharing data and materials.

service systems [4]. To date, prevention programs show mixed effects for females and males, and there are, as yet, no clear guidelines for what prevention program components are the most effective by gender. However, a number of female-specific programs shown to be effective at promoting health outcomes often include elements that emphasize empowerment and relationship skills. The current pilot study presents the feasibility and preliminary effectiveness of a female-specific prevention program that integrates empowerment, relational and cognitive behavioral skills. The program was developed for an indicated prevention population of female youth in contact with the justice system at moderate or high risk of recidivism, and was designed to work smoothly with court operations to promote sustainability.

## Adolescence as critical intervention point in preventing substance use

Adolescence is a significant turning point for developing sustained, problematic substance use [5–7]. While most adolescents will not go on to develop substance use disorders, those who begin using substances early and show patterns of moderate to heavy use during this time are at higher risk for developing adult substance use disorders [8]. The prevalence of use jumps sharply as youth transition to adulthood, between the ages of 13 and 18 [6]. Efforts to prevent substance use disorders are also likely to prevent general delinquency [9, 10] and aggression [11], making these services a good investment for both health and justice systems [11, 12].

## Cognitive behavioral interventions and gender

Cognitive behavioral approaches employ a shared framework for teaching skills related to managing emotions, challenging negative thoughts and problem-solving [13]. This approach has the strongest evidence of effectiveness for preventing and reducing substance use disorders [14–15], reducing adolescent aggression [16] and preventing adolescent delinquency [17]. In a meta-analytic review of substance use treatment programs for adolescents, Vaughn & Howard [14] rated cognitive behavioral therapy groups in the highest category for evidence (effect sizes greater than 0.20 in highly controlled designs). However, a nine-year follow-up study of substance use prevention programs using a CBT skills-based approach found that long term effects from these programs were only sustained for males [18]. Further, an earlier review of 47 general prevention programs by Fagan & Lindsey [19] found that 51% of programs varied in outcomes by gender. However, no consistent patterns emerged by program type that could account for these differences. A review of the delinquency intervention literature [20] found similarly mixed effects between genders with inconsistent patterns. To date, little is known about why programs for youth involved in the juvenile justice system demonstrate different outcomes by gender, but more effective approaches are clearly needed [19, 21].

## Female-specific programming

Researchers and advocates have called for female-specific programs that blend both relational and cognitive behavioral approaches [22] given the strong evidence for CBT but inconsistent results for females. Differences in the etiology of substance use disorders also support the need for gender-specific models. Compared to males, females' relationship with parents and conflict in the home is more strongly linked to substance use [23, 24]; aggressive behavior [25–27]; and delinquency [28]. Poor outcomes associated with family conflict are especially likely when females have higher sensitivity to emotional cues [29–33] and less assertive coping styles [23].

The predictors of substance use disorder for adolescent females suggest patterns of risk that may result from poor self-concept and low self-efficacy [34]. Consequently, effects of prevention programs for females are likely to be enhanced if they included content focused on building up perceived and actual efficacy. Indeed, examples of successful female-specific

programming for other health targets suggest relationship-based programs focused on empowerment, referred to as social empowerment programs (SEP), are particularly well-received by females and are becoming increasingly popular in domestic and global health [35, 36]. These programs typically include educational sessions on physical and emotional safety, relationship building, social awareness, and may include mentoring, problem-solving and assertiveness skills. SEPs can be effective in improving sexual health, HIV prevention and reduced domestic violence for adult females [37–39], and so there is reason to expect this approach would benefit adolescents as well. However, little is known about the effect of a standalone SEP approach for adolescent substance use prevention. Some mixed findings in the literature, particularly null effects in high needs populations [35], suggest SEP may not be effective alone and a blend of empowerment and CBT in program design may yield more robust effects for indicated prevention programs.

## Implementing prevention programs with court-based staff

Training juvenile probation officers to deliver prevention programming presents possible challenges. It is not clear that probation officers, hired and trained for a traditionally compliance-oriented job, can be feasibly trained to deliver second tier prevention services. The approach also raises some larger concerns about the wisdom of centering therapeutic services within courts, including the risk of retaining youth in court services who could otherwise be diverted into community services [40], and subjecting youth who are noncompliant with program services to higher levels of scrutiny [41]. Approaches using probation as the facilitators of prevention programming would need to be thoughtful about separating those roles. If done successfully, however, youth could be expected to benefit from a probation workforce more attuned to principles of positive youth development and therapeutic skill building [42]. Most studies of interventions involving court-referred youth study the impact of interventions delivered by providers external to court operations (e.g., [43, 44]). Very few published studies examine the feasibility of training internal court services staff (e.g., probation officers) to deliver interventions unless these approaches are framed as enhancing supervision services [45]. However, probation officers trained to conduct prevention services may be more likely to incorporate this framework into other areas of work, including their probation supervision approach [46, 47]. Further, using internal staff to provide prevention services may save costs by reducing the need for external contracting and increasing access to services.

## The current study

Accordingly, we present feasibility and preliminary outcomes of a pilot study that included a controlled trial of a female-specific substance use disorder and delinquency prevention program. The program was designed to work sustainably within a juvenile court setting using existing staff (juvenile probation officers and contracted community providers) as the facilitators. To support usability, the program design team included probation officers along with a sexual abuse and trauma expert, a clinical psychologist, a justice programs quality assurance manager, and a youth behavioral health implementation researcher.

The program development process used a co-design, participatory approach [48]. Co-design is a method of program development that engages the intended systems in the development process to ensure fit and sustainability. The process views sustainability to be equal in value to effectiveness. The development process thus attempts to engage all relevant local expertise in order to build capacity around existing systems. Consequently, in the present study, the development process began with a review of an existing CBT program already operating in the local juvenile courts [17, 49] to determine which elements aligned with the

| Week | Original |
|------|----------|
| 1 | Engagement |
| 1 | CBT Triangle |
| 2 | Five Major Feelings |
| 2 | Feelings Thermometer and Coping |
| 3 | Coping Skills and Emotion Surfing |
| 3 | Thought Disortions |
| 4 | Helpful Thoughts |
| 4 | Emotion and Thoughts Review |
| 5 | Goal Setting |
| 5 | Mid Session Review |
| 6 | Problem Solving |
| 6 | Friendship and Peer Refusal Skill |
| 7 | Friendship and Expressing Feelings Skill |
| 7 | Family and Dealing with Anger Skill |
| 8 | Family and Difficult Conversations Skill |
| 8 | Relationship and Understanding Feelings Skill |
| 9 | Gender Bias and Assertiveness Skill |
| 9 | Relationships and Asking for Help Skill |
| 10 | End of Session Review |
| 10 | Graduation |

**Fig 1. Program components.**

principles of female development and empowerment and where adaptations should be made. The decision to include probation officers as facilitators was largely guided by the courts' prior success in using probation officers as facilitators for a different CBT-based program [17]. The state quality assurance manager for this program served as a lead along with the first author in bringing together courts to discuss these adaptations, subsequently adding more content area expertise as the project evolved. The resulting program, Girls Only Active Learning (GOAL), looks significantly different from the original CBT program but has earned significant buy in and support from the state as a result of the development process (see Fig 1).

GOAL is a 20 session, 10-week program for female youth that includes a parent education component. The CBT elements are introduced early in the program with sessions 1–10 focused on learning and practicing emotion identification and coping skills, cognitive

restructuring, goal-setting, and assertive and problem-solving skills. Empowerment language is integrated throughout with an emphasis on "personal power." This is particularly used in the latter half of the sessions (10–19) where participants discuss and practice using skills in relationship-based scenarios related to family, peer, and romantic relationship contexts. The theory of change behind the program is based on a social development model, theorizing that enhancing the capacity of adolescents to adopt healthy responses to relational conflict in these domains while promoting a future orientation will lead to fewer opportunities for conflict and substance use [50]. The home environment of the participants is addressed through weekly text messages to parents (or current guardian) that describe the weekly program topic along with brief psychoeducation about healthy female development and suggestions for reinforcing skills being learned in group.

The goal of this study was two-fold. First, we examined the feasibility of co-designing and embedding a new prevention program within juvenile courts and assessed feasibility by tracking the number of implemented programs, the individual attendance rates, and acceptability using qualitative feedback from facilitators, program participants, and parents. Second, we examined the preliminary effectiveness of the program as compared to usual care through 3- and 6-month assessments of primary and secondary health and behavior outcomes.

## Method

### Participants

Eligible participants included 136 females referred from juvenile probation programs from five counties between September 2014 and December 2015. Of the eligible sample, $n = 101$ provided consent and $n = 87$ completed at least one survey over the three data collection waves. All participants were female, with the following racial and ethnic groupings: White (47%), mixed ethnicity (26%), Black (8%), Latina (7%), American Indian (2%) or other (9%), which is roughly equivalent to the distribution of ethnicity of youth involved in juvenile courts in Washington State. Participants were between 12–17 years old; $M[SD] = 15.15[1.72]$. Inclusion criteria were: identified as female, were on probation, and had at least one of the following risks: moderate family strain/conflict; exposure to violence; poor school achievement; or antisocial peer involvement as measured by the court risk and needs assessment used in all juvenile courts in the state (Positive Achievement Change Tool, PACT) [51]. Youth were excluded if they had a diagnosable substance use disorder or serious emotional disturbance as these needs require more intensive interventions.

### Procedures

This study employed a quasi-experimental design to assign participants to GOAL or services as usual by offering GOAL a limited number of times in each site and referring all eligible participants within this service window to GOAL and all eligible participants outside of this service window to usual programs, as described in more detail below (also see Fig 2). To work smoothly with the operations of the juvenile court, youth were not required to enroll in the research study in order to access GOAL. All eligible youth were referred to GOAL automatically when the program was available and to services as usual when it was not. Research recruitment occurred after enrollment in GOAL or services as usual.

Research eligibility was determined by the court probation counselors at each site who approached eligible participants about the study using a script and information sheet. Each site also designated a study liaison who reviewed eligibility and reached out to Probation Officers when an eligible youth was on their caseload. Once a probation officer received permission to share contact information from the youth, they securely faxed this information to the research

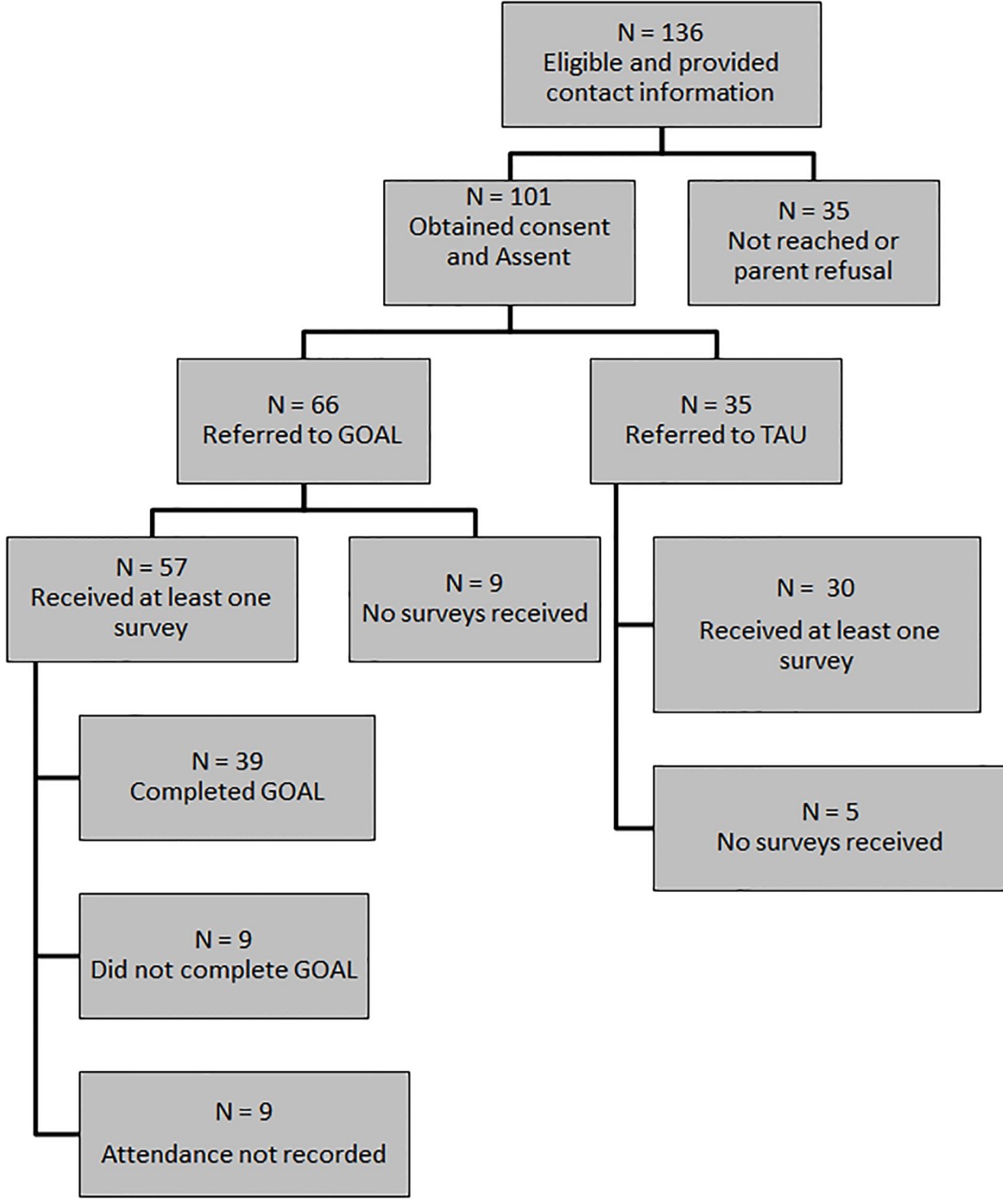

**Fig 2. Consort diagram.**

group who followed up with caregivers and youth to explain the study and solicit assent/consent over the phone. Youth provided assent for their own participation and parents provided verbal consent for their child to participate and to provide feedback on weekly parent information messages. Participants had the option of receiving surveys over email, text, or through

regular mail. The procedures were approved by the University of Washington Institutional Review Board.

Youth were assigned to either GOAL or services as usual based on the availability of the GOAL program which was strategically only offered twice a year in each court. Courts were instructed to refer all study eligible youth to GOAL during the period of GOAL enrollment. Over the study time period, the courts referred $n = 123$ youth to GOAL. Youth receiving treatment as usual were referred to services based on eligibility for other court and community programs already available to court-involved youth. For both GOAL and usual services, referral from probation officers was considered part of the court order and the services was considered completed if a youth did not miss more than three sessions. We did not ask courts to provide the study with information about the total number of eligible youth receiving services as usual.

Youth were surveyed at three points during the study: baseline, 3 months, and 6 months from study enrollment. These timeframes were established to assess pre-program, immediate post-program and sustained effects for those participating in GOAL compared to usual services. Parents with a daughter enrolled in GOAL were surveyed once after the program ended (second wave) to assess acceptability. Youth and parents were given the option to complete the surveys online or by mail. Youth received $10, $15 and $20 for returned surveys. Parents received $10 for returned surveys.

## GOAL implementation

Group facilitators were probation officers (50%), contracted-counselors (25%) and community youth workers (25%) trained by the study team to conduct the GOAL groups. Each site recruited individuals who fit the criteria below, and were typically currently delivering programs through the court or with court-referred youth in the community. A minimum of two facilitators were trained for each juvenile court. To be eligible, facilitators had to have either previous experience leading skills-based treatment groups with justice-involved youth, or previous experience working in therapeutic environments with adolescent youth. All probation officers in the study had previous experience leading an existing court-based therapeutic group, Aggression Replacement Training (ART). Other facilitators had a mix of experience with either facilitating ART or other therapeutic groups or services. A master's degree or two years post-bachelor equivalent experience was required to facilitate groups. In half of the sites (3), programs were conducted in rooms in the juvenile court facilities, in the other half (3), programs were conducted offsite in community settings.

Training was provided by two PhD-level, trained mental health clinicians with experience conducting therapeutic groups with adolescents. The training included 10 hours of active instruction and role play over two days and weekly consultation calls through the first round of facilitating the program (10 consultation calls). Throughout implementation of GOAL, facilitators completed self-assessment forms for each session on at least weekly basis. The forms assessed adherence to the intervention manual, and challenges and successes in implementation. The consultation calls focused on reviewing self-assessment forms, problem-solving issues that arose in implementation, and planning for upcoming sessions. Finally, facilitators were asked to videotape two sessions, which were reviewed by the trainers.

## Measures

**Measures of feasibility.** Feasibility was measured by rate of program completion and facilitator fidelity. Program completion was compared to a benchmark of program completion for other court-based services and the number of youth enrolled for each group. Fidelity was measured via facilitator self-report. For each session, one of the facilitators reported on "how

closely [they followed] the written curriculum (the education, activities, order/sequencing, discussion prompts, etc.)" on a scale of 1 = Exactly, 2 = Mostly, missed a couple of things, 3 = Only covered a few areas, and 4 = Didn't really follow it at all. They were also asked to describe and explain any modifications that were made via open-ended responses.

**Measures of acceptability.** Acceptability was measured with a study-developed satisfaction survey with Likert scale items and open-ended responses for youth participating in GOAL and their parents. Parent satisfaction was measured with six scaled items developed by the researchers with a range of 1–5 (strongly agree) to measure parents' sense of being informed about the program and whether the program helped their child. Youth satisfaction was assessed with six scaled items with a range of 1–7 (strongly agree) developed by the researchers which included items about general satisfaction, whether the respondent learned new skills and whether the respondent felt treated with respect, alpha = 0.72. Satisfaction surveys were obtained from parents and youth at 3 months.

**Substance use and consequences.** Substance use and consequences were measured with items taken from the Drug Involvement Scale for Adolescents (DISA) [52]. DISA was developed to assess the context and consequences of substance use. Use is measured on a 6-point scale (Not at all–Every day) for the last month and includes 12 controlled substances. Consequences of substance use is measured on a 7-point scale (Not at all–Six or more), including missing an assignment in class, being arrested, stealing, damaging property, late for work/class. Items were summed to create a frequency score across all substance types. Item reliability was good for baseline and 6-month follow-up (alphas of 0.87 and 0.83, respectively).

**Risk behavior.** Risk behavior was measured with the Risky Behaviors subscale of the High School Questionnaire [53], a reliable measure of youth offending. Youth responded to four questions on a 7-point scale (Not at all–Six or more) for the past month. Items include getting in a physical fight, threatening to hurt others, shoplifting, and hitting something. The risk behaviors were summed to create a frequency score. In the current study, item reliability was acceptable for baseline and 6 months (alphas of 0.70 and 0.73, respectively).

**Family climate.** Family climate was measured with the Communities that Care Youth Survey (CTCYS) [54]. This 8-item survey includes questions about house rules, parental support, and conflict among family members on a 5-point scale (Almost never–All of the time). Family climate in the home is predictive of later problem substance use and delinquency. In the current study, item reliability was acceptable for baseline and 6 months (alphas of 0.84 and 0.73, respectively).

**Emotion dysregulation.** Emotion regulation was measured with the Difficulties in Emotional Regulation Scale (DERS) [55]. The DERS assesses perception and beliefs about difficulties in emotional regulation and beliefs about emotional control. It is scored on a 5-point scale (Almost never–Almost always). In the current study, item reliability for the total scale was high for baseline and 6 months (alphas of 0.93 and 0.94, respectively).

**Quantitative data analyses and missing data.** Missing data within surveys was low, but there was notable missingness between waves of data collection, with only 26 youth (30%) completing all three waves. All but 7 youth completed one survey at baseline and/or 3 months (92%). Missingness was highest at the third wave (54%), compared to wave 1 (24%) and wave 2 (32%). The youth completing data only at 6 months were youth the study team had the most difficulty engaging and it cannot be assumed that these 7 youth had data missing at random. Consequently, these youth were removed from the outcome analyses, resulting in an analytic sample of $n = 80$. Scores were then analyzed with linear regression models using full information maximum-likelihood estimation using Mplus7 to estimate over missing data [56].

## Results

### Descriptive information

At the time of enrollment in the study, 57% of the sample had used an illicit substance in the past 30 days and 41% reported some negative consequence as a result of use. Emotion dysregulation was comparable to a population-based sample of high school students, $M[SD]$ = 2.74 [0.65]. Nearly half (45%) of youth reported having family conflict during "at least half" of their parent interactions. Participants in the comparison condition (i.e., services as usual) accessed a variety of services in the treatment as usual (TAU) condition. Twelve (50%) participants declined to respond to the question. Of the respondents, 4 received no services (14%), 5 received substance use disorder treatment (17%), 2 received ART (7%), 2 received mental health counseling (7%), and one received a youth development program (3%). Participants in the two conditions were balanced in age, race/ethnicity and measured baseline characteristics, confirming the design for condition assignment was adequate in balancing observed characteristics.

### Feasibility

Courts were largely able to meet the target group size with an average of 8 youth enrolled per group (range 6–13). Of those referred to GOAL and the study ($n$ = 66), 9 youth (14%) did not complete the program, which outperforms the benchmark rate for youths' typical attendance in court-based group programming in Washington State (36% non-completion). A one proportion test of significance found this to be a significant improvement in retention when compared to the existing rate, $z$ = 3.89, $p < .0001$.

Across all sessions and facilitators, the mean self-reported therapist fidelity was 1.64 (range = 1 to 2), indicating that facilitators followed the manual "exactly" or closely. A review of the qualitative descriptions and explanations for modifications suggested that the modifications were minor, and consisted primarily of adaptations (e.g., changing examples to enhance relevance for females), or omission of a session component (e.g., journaling).

### Acceptability

**Parent surveys.**   Parent acceptability of the weekly messages and of the program was high overall. However, the response rate for the parent surveys was 43% (25/57), so responses may indicate some bias towards extreme or favorable views. Responses from the six questions ranged from a mean of 3.86 to a high of 4.50 on a 5-point scale. On average, parents strongly agreed that they would recommend the program to other parents, $M[SD]$ = 4.5[0.63]. The updates were uniformly seen as helpful, $M[SD]$ = 4.21[0.75]. Parents also noted the value of the program in open-ended responses: "The weekly updates as to what would be addressed so I could speak to my teen about it. The updates were very thorough." "The concepts presented are directly relevant to the challenges [my daughter] is facing."

**Youth surveys.**   Just over half of the youth referred to GOAL completed satisfaction surveys, $n$ = 33 (58%). Those who responded had slightly lower reported substance use than youth who did not complete satisfaction surveys, $M[SD]$ = 14.31[3.44]; $M[SD]$ = 16.71[4.34] as well as lower reported risk behaviors, $M[SD]$ = 7.56[4.79]; $M[SD]$ = 8.52[6.12]. The average response to each of the six questions assessing GOAL satisfaction were "agree" or higher. The item with the most variation in response was whether the program helped, $M[SD]$ = 5.52 [1.50], with one youth reporting "strongly disagree" and another reporting "somewhat disagree" in addition to 75% reporting "somewhat agree" or higher. Responses included: "Getting to know my emotions more and getting to know how other people react to things and getting

**Table 1. Estimated means and outcome effects for GOAL and TAU at 3 and 6 months.**

| | Baseline | | | 3 Months | | 6 Months | | Intervention Effects (3 mos) | | | Intervention Effects (6 mos) | | |
|---|---|---|---|---|---|---|---|---|---|---|---|---|---|
| | N | Mean | SD | Mean | SD | Mean | SD | B | SE | β | B | SE | β |
| Substance Use | | | | | | | | | | | | | |
| GOAL | 54.00 | 15.75 | 0.70 | 14.77 | 0.63 | 14.83 | 0.69 | -1.98 | 1.03 | 0.15 | -2.28 | 1.27 | 0.40 |
| TAU | 26.00 | 16.66 | 0.85 | 17.15 | 1.00 | 18.56 | 0.90 | | | | | | |
| Behavior | | | | | | | | | | | | | |
| GOAL | 54.00 | 7.58 | 0.64 | 6.69 | 0.66 | 5.37 | 0.53 | -1.08 | 1.06 | 0.48 | -1.99 | 0.80 | **0.84** |
| TAU | 26.00 | 8.62 | 1.20 | 7.90 | 0.98 | 8.49 | 1.04 | | | | | | |
| Dysregulation | | | | | | | | | | | | | |
| GOAL | 54.00 | 2.74 | 0.09 | 2.63 | 0.11 | 2.72 | 0.14 | 0.00 | 0.14 | 0.18 | -0.01 | 0.19 | 0.10 |
| TAU | 26.00 | 2.62 | 0.17 | 2.59 | 0.16 | 2.69 | 0.20 | | | | | | |
| Family climate | | | | | | | | | | | | | |
| GOAL | 54.00 | 2.78 | 0.13 | 2.48 | 0.11 | 2.49 | 0.14 | -0.28 | 0.19 | 0.31 | -0.15 | 0.50 | 0.10 |
| TAU | 26.00 | 2.89 | 0.23 | 2.78 | 0.19 | 2.65 | 0.18 | | | | | | |

Notes. **Bold** = p < .05. For parsimony, estimated baseline means are taken from the three month models. Baseline means for the six month models differed negligibly from the three month models and are available from the authors. Effect sizes were calculated as the mean difference of the change by condition divided by the standard deviation of the dependent variable for each model.

to know their personality!" "How we got to do the talking, not just the instructors." "I wasn't judged negatively." "I liked talking to other youth with similar problems." When asked what could be improved, youth reported nothing ($n = 8$), listening more ($n = 5$), and adding more skills "because the skills helped a lot."

**Substance use.** The linear model predicting substance use at 3 months allowed age to correlate with baseline substance use and included age and baseline substance use as covariates, *CFI/TLI* = 0.99/0.99, *SRMR* = 0.06, *RMSEA* = 0.03. GOAL was negatively associated with substance use at 3 months, *M[SE]* = -1.98[1.03], *p* = 0.06, with a small effect, β = 0.15, but did not reach statistical significance (see Table 1).

The best fitting model for predicting substance use at 6 months included age and substance use at baseline as covariates predicting substance use at 6 months, *CFI/TLI* = 0.92/0.86, *SRMR* = 0.08, *RMSEA* = 0.11. GOAL was negatively associated with substance use, *M[SE]* = -2.28[1.27], *p* = 0.07, with a medium effect, β = 0.40, and did not reach statistical significance.

**Risky behavior.** The model predicting behavior at 3 months allowed age to correlate with behaviors at baseline and included age and baseline behavior, *CFI/TLI* = 0.91/0.87, *SRMR* = 0.06, *RMSEA* = 0.06. GOAL was negatively but not significantly associated with behavior at 3 months, *M[SE]* = -1.08[1.06], *p* = ns, with a medium effect, β = 0.48.

The model predicting risky behavior at 6 months allowed risky behavior at baseline and 3 months to correlate, along with age as covariates in the model, *CFI/TLI* = 0.97/0.92, *SRMR* = 0.06, *RMSEA* = 0.08. GOAL was negatively and significantly associated with risky behavior at 6 months, *M[SE]* = -1.99[0.80], *p* < .05, with a strong effect, β = 0.84.

**Emotion dysregulation.** The model predicting emotion dysregulation (ED) at 3 months allowed age to correlate with ED at baseline and included age and ED at baseline as covariates, *CFI/TLI* = 0.99/0.99, *SRMR* = 0.05, *RMSEA* = 0.04. GOAL was not significantly related to ED at 3 months, *M[SE]* = 0.003[0.14], *p* = ns, and had a small effect, β = 0.18.

The model predicting ED at 6 months allowed age and ED at baseline to correlate and included age and ED at baseline as covariates, *CFI/TLI* = 0.99/0.99, *SRMR* = 0.05,

*RMSEA* = 0.04. GOAL was not significantly associated with ED at 6 months, *M[SE]* = -0.01 [0.19], *p* = ns, and had a small effect, β = 0.10.

**Family climate.** The model predicting family conflict (FC) at 3 months included baseline FC as a covariate, *CFI/TLI* = 1.0/1.0, *SRMR* = 0.02, *RMSEA* = 0.00. GOAL was negatively but not significantly associated with family climate at 3 months, *M[SE]* = -0.28[0.19], *p* = 0.14, and had a medium effect, β = 0.31.

The model predicting family climate at 6 months allowed age to correlate with family climate at baseline and included age and baseline family climate as covariates, *CFI/TLI* = 1.0/ 1.12, *SRMR* = 0.004, *RMSEA* = 0.00. GOAL was negatively but not significantly associated with family climate at 6 months, *M[SE]* = -0.15[0.19], *ES* = 0.10, *p* = ns, and had a small effect, β = 0.10.

## Discussion

The purpose of the study was to assess the acceptability, feasibility, and preliminary effectiveness of a female-specific, CBT program for reducing substance use risk and delinquency for youth in contact with the justice system. Taken together, results from this quasi-experimental pilot study suggest that GOAL is a promising intervention for targeting the indicated treatment needs of this group.

Study findings suggest the program can be feasibly implemented by probation and court-contracted group facilitators who have previous experience running therapeutic groups. The fidelity scores indicated high adherence to the program with only minor adaptations and attendance exceeded female attendance rates for other court-based programming. Feedback from parents and youth who responded to the program satisfaction surveys indicated very high acceptability but these appear to underrepresent youth at higher risk for substance use and delinquency. Consequently, it is not clear that higher risk youth viewed the programs as helpful so results can only be assumed to hold for females at moderate, but perhaps, not highest risk for ongoing substance use. GOAL was also very well-received by a little over half of the youth responding the study survey, which represents just under 30% of all of the youth who participated in GOAL during the study timeframe. Missing responses did not appear to be random, as youth not responding to the satisfaction survey had more risk factors at the baseline assessment. Consequently, the program appears to be a good fit for a number of youth but this may not hold as youths' needs increase.

Taken together, the female-specific program seems well-suited for courts with either court-contracted staff or probation officers who are already familiar with running therapeutic skills groups. The model presumes a baseline level of therapeutic knowledge and skill that may not be as readily present in other juvenile court systems. Results regarding the preliminary effectiveness of GOAL were promising, with notable findings for reducing risk behaviors associated with delinquency. Specifically, there was a statistically significant, large effect for reduced risk behaviors at the 6-month follow up for the GOAL group compared to controls. For 3- and 6-month substance use, the mean differences were in the expected direction, and the results showed trend-level effects (*p* < .10) for GOAL compared to TAU. For 3-month risky behavior, 3- and 6-month family climate, and 3- and 6-month emotional dysregulation, no significant differences were observed between the intervention conditions. The findings suggest that GOAL may be effective in reducing the primary targets of delinquency and substance use but the lack of observed change in emotion regulation suggests the mechanisms of change may be different than hypothesized.

The mechanisms of change for substance use and delinquency prevention interventions are not well-understood. In the social development model, the presumed effects lie in the

success of the model in changing the social and internal incentives to participate in prosocial rather than antisocial activities [51]. The theorized mechanism of effect of a CBT approach is the expected increase in consequential thinking in order to help youth understand that short term choices have long term consequences. This is then expected to build youth motivation to participate in prosocial activities and mitigate the influence peers will otherwise have on decision-making in the middle adolescent years. Consequently, the model assumes that success in the intervention effects should be mediated by the youth's successful use of these skills but this was not observed in our study. We did not measure problem-solving skills with a standardized measure, although qualitative responses from youth indicated some support for practicing problem-solving skills in role plays. This provides some qualitative support that the rehearsal and building of cognitive and behavioral skills had some effect on outcomes.

An alternative model of substance use and delinquency intervention using mindfulness-based intervention (MBI) proposes that decisions to engage in risk behaviors can be mitigated by an increased ability to resist distressing emotions [57]. Mindfulness-based interventions (MBI) attempt to reduce negative thoughts and emotions by helping the individual focus on present sensation [58, 59]. Interventions employing mindfulness in isolation or as part of a multicomponent curriculum are based on the theory that individuals are building their ability to manage distress through direct manipulation of their physical states, which leads to less perceived urgency and impulsivity [60]. This capacity is also observed to affect other areas of health and behavior as individuals gain more insight into the relationship between mind and body functioning [61]. We would expect to see some change in emotion regulation skills through this mechanism of action in GOAL, which did include skills training around mindfulness and emotion regulation. The lack of an observed effect within or between groups in the study could be due to insufficient dosage, lack of adequate skills transfer or lack of specificity in measurement. Also, the scale used to measure emotion regulation, the Difficulties in Emotional Regulation Scale (DERS), is a multicomponent scale. While the total score demonstrates the best predictive capacity in psychological studies [62], it is possible that the intervention exerted effects on more specific skill areas than detected by the total DERS score. Our sample was not sufficiently large to power analyses to test this theory and would need to be replicated by a larger study.

## Limitations

This pilot study was limited by a relatively small sample size for detecting effects in a sample in which the majority of youth in the comparison condition were participating in other therapeutic programs. As a pilot, the study had limited resources to track down youth who did not respond to the surveys after email and phone contact. Because the study was focused on rolling out a codesigned program across multiple sites, we focused resources on maximizing the reach of the program rather than improving the survey response rate. While the sampling can be considered missing at random, and robust estimators were used to estimate over missingness, it is not possible to completely control for bias that may have impacted the outcomes.

The current evaluation will need to be replicated with a larger sample. A larger study will also provide sufficient power to model nested effects for youth outcomes within facilitator characteristics. Youth in the study were assigned to GOAL or control conditions using a staggered model that appeared to be successful in balancing group differences but was not random. Further, youth only came to the attention of the study team following referral from the courts and potential differences in referred and non-referred youth are not available to confirm the generalizability of the findings to all potentially eligible participants. The study recruited fewer

youth from the treatment as usual group compared to GOAL, and we cannot be sure that this did not introduce some bias into the sampling.

While parent acceptability was high, the results are limited to parents being willing to respond to the parent survey. Parents and youth not responding to the survey may have been less motivated to respond because they found less value in the program. Replication of parent acceptability is needed with a higher participant response.

Finally, our measure of fidelity only assessed adherence to intervention elements via facilitator self-report. Previous studies have shown that front-line providers are accurate reporters of their adherence, and self-reported adherence significantly predicts response to intervention [63, 64]. Nevertheless, future research should incorporate other dimensions of fidelity that may impact participant outcomes, such as facilitator competence. Similarly, self-reported delinquent behavior and substance use outcomes, while shown to be reliable indicators in other studies [62, 65], may underestimate the true incidence of these behaviors and could reflect bias in reporting.

Missingness between waves of data collection was notable and while full information maximum likelihood is robust up to 50% missingness [66], replication is needed to ensure study results are not due to biases introduced by the estimated models.

## Conclusions

Overall, this pilot demonstrated the feasibility and acceptability of a substance use and delinquency prevention program for females involved with the justice system. The pilot additionally demonstrated promising results for reduced delinquency compared to other, non gender-specific services and trend level effects for reduced substance use. The study failed to find effects on reducing overall emotion regulation capacity, one of the hypothesized mechanisms of action. Additional study is necessary to identify whether the effects were limited to lower risk youth and if the program theory of change is mis-specified or if the measures used in the study were not sensitive enough to detect the hypothesized mechanisms of change.

## Acknowledgments

We are grateful for the participation of the site facilitators, youth and parent respondents for this study, as well as the project support from Kaely Wickham, Rita Olson, Lynette Walker, Hyunjung So, Mallory Halbert, Kristin Vick and Emi Gilbert.

## Author Contributions

**Conceptualization:** Sarah C. Walker, Mylien Duong, Christopher Hayes, Lucy Berliner, Leslie D. Leve, David C. Atkins.

**Investigation:** Mylien Duong, Asia S. Bishop, Esteban Valencia.

**Methodology:** Sarah C. Walker, Mylien Duong, Leslie D. Leve, David C. Atkins, Jerald R. Herting.

**Supervision:** Sarah C. Walker.

**Writing – original draft:** Sarah C. Walker, Mylien Duong, Asia S. Bishop, Esteban Valencia.

**Writing – review & editing:** Leslie D. Leve, David C. Atkins, Jerald R. Herting.

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
