## [Decision Letter · Decision Letter 0]

12 Aug 2019

PONE-D-19-16985

A tailored cognitive behavioral program for juvenile justice-referred females at risk of substance use and delinquency: A pilot quasi-experimental trial

PLOS ONE

Dear Dr. Walker,

Thank you for submitting your manuscript to PLOS ONE. After careful consideration, we feel that it has merit but does not fully meet PLOS ONE’s publication criteria as it currently stands. Therefore, we invite you to submit a revised version of the manuscript that addresses the points raised during the review process.

We would appreciate receiving your revised manuscript by Sep 26 2019 11:59PM. To enhance the reproducibility of your results, we recommend that if applicable you deposit your laboratory protocols in protocols.io, where a protocol can be assigned its own identifier (DOI) such that it can be cited independently in the future. For instructions see: http://journals.plos.org/plosone/s/submission-guidelines#loc-laboratory-protocols

We look forward to receiving your revised manuscript.

Kind regards,

Thach Duc Tran, M.Sc., Ph.D.

Academic Editor

PLOS ONE

Journal Requirements:

1. Thank you for including your competing interests statement; "I have read the journal's policy and the authors of this manuscript have the following competing interests: S.C.W., L.B., C.H. and M.D. are listed as co-developers on a University based license for the distribution of the GOAL curriculum (patent pending)."

We note that you have a patent relating to material pertinent to this article. Please provide an amended statement of Competing Interests to declare this patent (with details including name and number), along with any other relevant declarations relating to employment, consultancy, patents, products in development or modified products etc. Please confirm that this does not alter your adherence to all PLOS ONE policies on sharing data and materials, as detailed online in our guide for authors http://journals.plos.org/plosone/s/competing-interests by including the following statement: "This does not alter our adherence to  PLOS ONE policies on sharing data and materials.” If there are restrictions on sharing of data and/or materials, please state these. Please note that we cannot proceed with consideration of your article until this information has been declared.

Reviewers' comments:

Reviewer's Responses to Questions

**Comments to the Author**

1. Is the manuscript technically sound, and do the data support the conclusions?

Reviewer #1: Yes

Reviewer #2: Partly

2. Has the statistical analysis been performed appropriately and rigorously? 

Reviewer #1: Yes

Reviewer #2: Yes

3. Have the authors made all data underlying the findings in their manuscript fully available?

Reviewer #1: Yes

Reviewer #2: No

4. Is the manuscript presented in an intelligible fashion and written in standard English?

Reviewer #1: Yes

Reviewer #2: Yes

5. Review Comments to the Author

Reviewer #1: Authors should describe the effect size results, especially since this is a small sample and the effect size results help interpret the effectiveness of the intervention.

Authors should review more literature relevant to female specific interventions--this is a growing field and placing this study within the existing literature can be done better.

Reviewer #2: This is a review of “A tailored cognitive behavioral program for juvenile justice-referred females at risk of substance use and delinquency: A pilot quasi-experimental trial.” As a licensed clinical psychologist who contracts to supervise psychology graduate students delivering an intervention in a youth court system I was initially very interested in this project. I think the topic of trying to test what is both effective and feasible for high at-risk youths like those in juvenile justice is an important goal but felt that this study as it is now presented fell short of providing information that is practically useful. I recognize that it is a pilot study, but nonetheless I have some concerns about both the way the information is presented in the manuscript and how the results would really provide actionable information to other researchers or court personnel. My comments below are mostly in order of the document.

The authors should consider updating terms like “substance abuse” and “dependence”.

This intervention seems like a lot of extra work for POs and they are not typically appropriately trained for this kind of thing. One also wonders about dual relationships. I think it really benefits the kids I work with to have a therapist who is not also dealing with their probation compliance. I actually argued against having a PO with a master’s degree in counseling provide the clinical services for this exact reason- you can’t expect to wear both hats and have it go well. For what it’s worth, the immediate reaction of every (psychology) colleague I mentioned the counselor-PO thing to was essentially “oh my God, no” so I wonder about the wisdom of this intervention, even though the authors were careful to select qualified people and had buy-in from the local system.

How many kids were in actual parental custody?

On p. 5 there is a citation listed as (author, year) rather than the numbering scheme used elsewhere.

Is there a citation for “probation officers who are familiar with principles of prevention are likely to incorporate this framework into other areas of work, including their probation supervision approach and participation in organizational policymaking” (p. 5)?

The authors claim that concerns about whether POs can or should deliver the intervention are “outside the scope of the current study” and I would insist that they are not. Both the competence of these non-mental health professionals, the effects of their having an apparent dual relationship with the participants (counselor and PO), and how well this worked on a macro level (did they have time to do this and the rest of their job well?) seem like pretty important points as far as the utility of this study.

The n of 87 doesn’t seem to match the CONSORT chart. I guess that is everyone who did at least one survey? I wonder if the authors should be more clear about how many youths did (or did not) do which aspects of the study up front.

How were emotion dysregulation and family conflict measured (p. 8)? Means are given before measures info.

Does “the court risk and needs assessment” have a name (p. 8)?

Why were so many more girls referred to GOALS than TAU?

The “participant self-report at 3and 6 months follow up” on p. 9 is 1) confusing as described and 2) there is a space needed between 3 and ‘and’ . 12 participants is 50% of what, exactly? And what questions were being asked?

Why the mix of group leaders? The intro is written like it will be POs but then it is only half POs. And I know this is a small sample but were there any effects of leader type? And POs who have 2 years of groups/MH experience and/or a master’s degree seems like a pretty special bunch. How does this program translate to any other jurisdiction?

The way feasibility was measured here is not really in line with the spirit of determining actual feasibility. People completed the program is certainly a component of that, but so are things like whether it was overly burdensome on staff, etc. Was group leader satisfaction assessed?

Why was there so much missing data? This is a tough population, but I’m still surprised at how high it is. Do the authors have an explanation?

How was the survey given to the participants? There isn’t really a procedure section for the surveys, and the GOAL group itself could be explained in more detail in a paper ostensibly about how its implementation went. Was acceptability collected at the same time as other data? How many youths actually provided this information? It’s less than half of parents and general follow-up with youths has a lot of non-responses so this should be specified.

Were kids mandated to attend treatment? What were the consequences of not attending or completing? What were to reasons for non-completion of poor attendance?

p.15- “each of the six question” question should be plural.

The youth response that an improvement would be “more food” suggests that food was provided but this is not mentioned anywhere.

I am wondering if this is an initial paper reporting some early results of a larger project, but in any case one wonders about statistical power in this small of a sample. Was there enough power to detect effects in the linear regressions with multiple IVs?

Something like a latent growth model might make more sense than separate linear models for two time points, though I suppose the sample is too small (and has too much missing data) for that to make sense.

The discussion overstates the findings a bit given the amount of missing data. The intervention doesn’t appear to have been unacceptable to the people who provided feedback, but that’s the minority of youths and parents so it’s hard to say how acceptable it really is.

In the discussion the authors should probably focus on effect sizes and more clearly discuss issues of sample size and statistical power rather than focus on significance.

If the authors are going to propose problem-solving as a potential mechanism, they should probably provide those (qualitative) data.

Similarly, suggesting that the facets of the DERS may have had more information than the mean of for all items (is that what the score is? Because that’s unusual, the measure has scoring guidelines) begs the question or why don’t the authors look at the facets?

The self-report nature of the outcome variables (for information that might easily be reported in a biased manner for a number of reasons) should also be mentioned as a limitation.

6. PLOS authors have the option to publish the peer review history of their article (what does this mean?). If published, this will include your full peer review and any attached files.

Reviewer #1: No

Reviewer #2: No

---

## [Author Response · Author response to Decision Letter 0]

26 Sep 2019

Response to Reviewers

Ref: PONE-D-19-16985 entitled “A tailored cognitive behavioral program for juvenile justice-referred females at risk of substance use and delinquency: A pilot quasi-experimental trial”

We thank the editor and both reviewers for the thorough read and helpful suggestions. We have made revisions to the manuscript to address each concern and provide details below explaining where changes can be found in the revised version. 

Editor:

1. Please ensure that your manuscript meets PLOS ONE's style requirements, including those for file naming. The PLOS ONE style templates can be found at: http://www.journals.plos.org/plosone/s/file?id=wjVg/PLOSOne_formatting_sample_main_body.pdf

Response: 

We have made formatting edits to comply with PLOS ONE’s style requirements. Thank you.

2. We note that you have a patent relating to material pertinent to this article. Please provide an amended statement of Competing Interests to declare this patent (with details including name and number), along with any other relevant declarations relating to employment, consultancy, patents, products in development or modified products etc. Please confirm that this does not alter your adherence to all PLOS ONE policies on sharing data and materials, as detailed online in our guide for authors http://journals.plos.org/plosone/s/competing-interests by including the following statement: "This does not alter our adherence to PLOS ONE policies on sharing data and materials.” If there are restrictions on sharing of data and/or materials, please state these. Please note that we cannot proceed with consideration of your article until this information has been declared. This information should be included in your cover letter; we will change the online submission form on your behalf. Please know it is PLOS ONE policy for corresponding authors to declare, on behalf of all authors, all potential competing interests for the purposes of transparency. PLOS defines a competing interest as anything that interferes with, or could reasonably be perceived as interfering with, the full and objective presentation, peer review, editorial decision-making, or publication of research or non-research articles submitted to one of the journals. Competing interests can be financial or non-financial, professional, or personal. Competing interests can arise in relationship to an organization or another person. Please follow this link to our website for more details on competing interests: http://journals.plos.org/plosone/s/competing-interests

Response: We revised our patent statement. We have a statement of invention filed with the National Institute of Drug Abuse and a license with our University but not a patent. We added a statement to the cover letter indicating which authors are listed as inventors for the federal grant and in the license. 

Response: We will upload and obtain a DOI for making the data available if the paper is accepted, thank you. 

Reviewer #1: 

4. Authors should describe the effect size results, especially since this is a small sample and the effect size results help interpret the effectiveness of the intervention.

Response:

Thank you. The effect sizes were reported and identified in the study as ES (Effect Size) statistics which are standardized regression coefficients (standardized beta) and interpreted the same as Cohen d (<.20 small, .30-.70 medium, >.80 large). We have now relabeled them so they can be more easily recognized as standardized beta coefficients (pp. 17-19). 

5. Authors should review more literature relevant to female specific interventions--this is a growing field and placing this study within the existing literature can be done better.

Response:

We appreciate this comment and added more description and references of female-specific programs relevant to our area of study in this section (pp. 5). 

Reviewer #2:

6. The authors should consider updating terms like “substance abuse” and “dependence”

Response: The terms “substance abuse” and “dependence” have been changed to “substance use disorder” or “problem substance use” in the revised manuscript. (e.g. pp. 2). 

7. This intervention seems like a lot of extra work for POs and they are not typically appropriately trained for this kind of thing. One also wonders about dual relationships. I think it really benefits the kids I work with to have a therapist who is not also dealing with their probation compliance. I actually argued against having a PO with a master’s degree in counseling provide the clinical services for this exact reason- you can’t expect to wear both hats and have it go well. For what it’s worth, the immediate reaction of every (psychology) colleague I mentioned the counselor-PO thing to was essentially “oh my God, no” so I wonder about the wisdom of this intervention, even though the authors were careful to select qualified people and had buy-in from the local system.

Response: 

We appreciate the challenges involved with having Probation Officers deliver therapeutically oriented services and have attempted to capture issues of acceptability and feasibility in the study. In the sites involved in the study, many of the PO’s were already providing group-based prevention programming. Using PO’s was one of the decisions made in the design process by the participatory team, which included system leaders. We have added this information to clarify (pp. 7). In addition, care was taken to ensure youth were not assigned to groups facilitated by their own PO and the intent was not to train PO as clinicians for their own probationers. We add this description in the text (pp. 6). 

8. How many kids were in actual parental custody?

Response:

All of the youth were in parental custody during the course of the study. 

9. On p. 5 there is a citation listed as (author, year) rather than the numbering scheme used elsewhere.

Response:

This was corrected, thank you (pp. 5).

10. Is there a citation for “probation officers who are familiar with principles of prevention are likely to incorporate this framework into other areas of work, including their probation supervision approach and participation in organizational policymaking” (p. 5)?

Response: 

Thank you for pointing out that this section could benefit from more literature support. We added two citations for this sentence: Charles Schwalbe’s 2012 study of the different approaches probation officers take in supervision with youth (social work oriented vs. compliance oriented) and Katherine Schwartz’s 2017 study looking at the use of motivational strategies to encourage youth success while on probation (pp. 6). 

11. The authors claim that concerns about whether POs can or should deliver the intervention are “outside the scope of the current study” and I would insist that they are not. Both the competence of these non-mental health professionals, the effects of their having an apparent dual relationship with the participants (counselor and PO), and how well this worked on a macro level (did they have time to do this and the rest of their job well?) seem like pretty important points as far as the utility of this study.

Response:

We clarified that the scope of the study was to assess acceptability and feasibility of the program, which we think bears on whether POs can be feasibly trained to deliver the intervention. (pp. 8).We added some language in the discussion to note that this does not mean this should be the first option for all or most youth. (pp. 20).

12. The n of 87 doesn’t seem to match the CONSORT chart. I guess that is everyone who did at least one survey? I wonder if the authors should be more clear about how many youths did (or did not) do which aspects of the study up front.

Response:

Thank you, we clarify that the sample included 136 girls of which 87 ended up completing at least one survey (pp. 8). 

13. How were emotion dysregulation and family conflict measured (p. 8)? Means are given before measures info

Response:

Thank you, we moved the description of baseline needs to the first paragraph in the Results section (pp. 15).

14. Does “the court risk and needs assessment” have a name (p. 8)?

Response:

We added the title of the assessment, the Positive Achievement Change Tool (PACT) to the description (pp. 9).

15. Why were so many more girls referred to GOALS than TAU?

Response:

We added language to clarify that as a real world research design, participants were not referred to an intervention after enrollment in the study. Youth were referred to programming and then recruited to be in the study. We add language to suggest that GOAL recruitment was likely higher because both the youth’s PO and GOAL facilitator was likely to give them info about the study, whereas only PO’s referred youth in the SAU condition (pp. 9). 

16. The “participant self-report at 3and 6 months follow up” on p. 9 is 1) confusing as described and 2) there is a space needed between 3 and ‘and’ . 12 participants is 50% of what, exactly? And what questions were being asked?

Response:

Thank you. In rereading this section, we agree that including information about the breakdown of services received in SAU is confusing as the section is intending to describe the group assignment process. We moved this information to the result section, and have corrected the typo (pp. 15).

17. Why the mix of group leaders? The intro is written like it will be POs but then it is only half POs. And I know this is a small sample but were there any effects of leader type? And POs who have 2 years of groups/MH experience and/or a master’s degree seems like a pretty special bunch. How does this program translate to any other jurisdiction?

Response:

We added language to clarify that each site independently recruited the staff to be trained to implement the program. The staff needed to meet the criteria and were typically individuals already delivering therapeutic programming through the court or in the community with court referred youth. (pp. 11).

18. The way feasibility was measured here is not really in line with the spirit of determining actual feasibility. People completed the program is certainly a component of that, but so are things like whether it was overly burdensome on staff, etc. Was group leader satisfaction assessed?

Response:

To provide a more complete picture of feasibility, we moved the self-reported facilitator fidelity into this section. (pp. 12, 15). Our team also did an in-depth qualitative study of facilitator feedback on the curriculum, but we do not have the space to report those results in this paper and they were consistent with the fidelity scores (e.g., the facilitator were able to implement and found the program, overall, to be helpful and useful). The reviewer comments are helping us recognize that readers of the paper may have a number of questions about the implementation of this program using probation officers. We attempt to address this in a number of places to clarify the limitations and possibilities of this approach (e.g. pp. 20). 

19. Why was there so much missing data? This is a tough population, but I’m still surprised at how high it is. Do the authors have an explanation?

Response:

The missing data largely reflects the funding level (R21) of the project as a pilot. We had limited funds to follow up with youth who did not complete the surveys after email/phone contact. Our goal was to obtain enough data to examine promising effects, but would need replication in order to confirm efficacy. We added a statement to this effect in the limitations section (pp. 23). 

20. How was the survey given to the participants? There isn’t really a procedure section for the surveys, and the GOAL group itself could be explained in more detail in a paper ostensibly about how its implementation went. Was acceptability collected at the same time as other data? How many youths actually provided this information? It’s less than half of parents and general follow-up with youths has a lot of non-responses so this should be specified.

Response:

Thank you for these prompts. We added more information on survey procedures and timing in the methods section under procedures (pp. 9-10) and measures (pp. 13). 

21. Were kids mandated to attend treatment? What were the consequences of not attending or completing? What were to reasons for non-completion of poor attendance?

Response:

We added language in the GOAL implementation section to clarify that referral to GOAL was similar to other programs referred through probation in that it was considered part of the court order (pp. 11). 

22. p.15- “each of the six question” question should be plural.

Response:

Thank you, this was changed. (pp. 17).

23. The youth response that an improvement would be “more food” suggests that food was provided but this is not mentioned anywhere.

Response:

Some facilitators of GOAL groups provided light snacks, but this was not considered essential to the program model. To reduce confusion, we omitted this feedback in the results (pp. 17).

24. I am wondering if this is an initial paper reporting some early results of a larger project, but in any case one wonders about statistical power in this small of a sample. Was there enough power to detect effects in the linear regressions with multiple IVs?

Response:

Thank you. As per the other reviewer comments as well, we make the reporting of effect sizes more clear and note the meaningfully large effects (e.g. pp. 17-19). We also more clearly note that this was a pilot study. (pp. 6).

25. Something like a latent growth model might make more sense than separate linear models for two time points, though I suppose the sample is too small (and has too much missing data) for that to make sense.

Response:

Yes, we agree, in this case the sample is much too small for latent growth modeling. 

26. The discussion overstates the findings a bit given the amount of missing data. The intervention doesn’t appear to have been unacceptable to the people who provided feedback, but that’s the minority of youths and parents so it’s hard to say how acceptable it really is.

Response:

Thank you, we edited the language to reflect the nuance in the findings, particularly in regards to the missing satisfaction data for youth (pp. 20). 

27. In the discussion the authors should probably focus on effect sizes and more clearly discuss issues of sample size and statistical power rather than focus on significance.

Response:

We agree and added that information (pp. 21, 22). 

28. If the authors are going to propose problem-solving as a potential mechanism, they should probably provide those (qualitative) data.

Response: We reported some of the qualitative feedback from the youth on the value of learning and practicing skills in the results section (pp. 17). We changed the term “broad support” to “some support” to more directly reflect the data reported in this paper (pp. 21). 

29. Similarly, suggesting that the facets of the DERS may have had more information than the mean of for all items (is that what the score is? Because that’s unusual, the measure has scoring guidelines) begs the question or why don’t the authors look at the facets

Response:

Our understanding (e.g., Hallion et al., 2018) is that the total score has the best psychometric properties for predicting psychological disorders. Our point in the discussion was intended to convey that the effect of the intervention may have had more specific effects not detected by the total score. We modified the language to clarify (pp. 22). 

30. The self-report nature of the outcome variables (for information that might easily be reported in a biased manner for a number of reasons) should also be mentioned as a limitation.

Response:We agree and have added language accordingly (pp. 24).

---

## [Decision Letter · Decision Letter 1]

14 Oct 2019

A tailored cognitive behavioral program for juvenile justice-referred females at risk of substance use and delinquency: A pilot quasi-experimental trial

PONE-D-19-16985R1

Dear Dr. Walker,

We are pleased to inform you that your manuscript has been judged scientifically suitable for publication and will be formally accepted for publication once it complies with all outstanding technical requirements.

With kind regards,

Thach Duc Tran, M.Sc., Ph.D.

Academic Editor

PLOS ONE

Additional Editor Comments (optional):

Reviewers' comments:

Reviewer's Responses to Questions

**Comments to the Author**

1. If the authors have adequately addressed your comments raised in a previous round of review and you feel that this manuscript is now acceptable for publication, you may indicate that here to bypass the “Comments to the Author” section, enter your conflict of interest statement in the “Confidential to Editor” section, and submit your "Accept" recommendation.

Reviewer #1: All comments have been addressed

Reviewer #2: All comments have been addressed

2. Is the manuscript technically sound, and do the data support the conclusions?

Reviewer #1: Yes

Reviewer #2: (No Response)

3. Has the statistical analysis been performed appropriately and rigorously? 

Reviewer #1: Yes

Reviewer #2: (No Response)

4. Have the authors made all data underlying the findings in their manuscript fully available?

Reviewer #1: Yes

Reviewer #2: (No Response)

5. Is the manuscript presented in an intelligible fashion and written in standard English?

Reviewer #1: Yes

Reviewer #2: (No Response)

6. Review Comments to the Author

Reviewer #1: (No Response)

Reviewer #2: (No Response)

7. PLOS authors have the option to publish the peer review history of their article (what does this mean?). If published, this will include your full peer review and any attached files.

Reviewer #1: No

Reviewer #2: No

---

## [Editor Report · Acceptance letter]

29 Oct 2019

PONE-D-19-16985R1 

A tailored cognitive behavioral program for juvenile justice-referred females at risk of substance use and delinquency: A pilot quasi-experimental trial 

Dear Dr. Walker:

I am pleased to inform you that your manuscript has been deemed suitable for publication in PLOS ONE. Congratulations! Your manuscript is now with our production department. 

With kind regards,

on behalf of

Dr. Thach Duc Tran 

Academic Editor

PLOS ONE